# Proinflammatory Responses in SARS-CoV-2 and Soluble Spike Glycoprotein S1 Subunit Activated Human Macrophages

**DOI:** 10.3390/v15030754

**Published:** 2023-03-15

**Authors:** Kim Chiok, Kevin Hutchison, Lindsay Grace Miller, Santanu Bose, Tanya A. Miura

**Affiliations:** 1Department of Veterinary Microbiology and Pathology, College of Veterinary Medicine, Washington State University, Pullman, WA 99164, USAsantanu.bose@wsu.edu (S.B.); 2Department of Biological Sciences, University of Idaho, Moscow, ID 83844, USA; 3Institute for Modeling Collaboration and Innovation, University of Idaho, Moscow, ID 83844, USA

**Keywords:** COVID-19, SARS-CoV-2, hyperinflammation, spike, S1 subunit, macrophages

## Abstract

Critically ill COVID-19 patients display signs of generalized hyperinflammation. Macrophages trigger inflammation to eliminate pathogens and repair tissue, but this process can also lead to hyperinflammation and resulting exaggerated disease. The role of macrophages in dysregulated inflammation during SARS-CoV-2 infection is poorly understood. We inoculated and treated human macrophage cell line THP-1 with SARS-CoV-2 and purified, glycosylated, soluble SARS-CoV-2 spike protein S1 subunit (S1) to clarify the role of macrophages in pro-inflammatory responses. Soluble S1 upregulated TNF-α and CXCL10 mRNAs, and induced secretion of TNF-α from THP-1 macrophages. While THP-1 macrophages did not support productive SARS-CoV-2 replication or viral entry, virus exposure resulted in upregulation of both TNF-α and CXCL10 genes. Our study shows that extracellular soluble S1 protein is a key viral component inducing pro-inflammatory responses in macrophages, independent of virus replication. Thus, virus- or soluble S1-activated macrophages may become sources of pro-inflammatory mediators contributing to hyperinflammation in COVID-19 patients.

## 1. Introduction

Severe acute respiratory syndrome coronavirus 2 (SARS-CoV-2) is the causative agent of coronavirus disease 2019 (COVID-19). Severely ill COVID-19 patients display lung tissue damage associated with cell death and pathologic inflammation [1,2], which are linked to enhanced pro-inflammatory cytokine and chemokine levels (e.g., TNF-α and CXCL10) [3,4]. These pathologies are compatible with a dysregulated inflammatory response characteristic of cytokine release syndrome or macrophage activation syndrome [5] and generalized hyperinflammation [6]. These patients often progress to respiratory failure due to complications from hyperinflammation and require mechanical ventilation. Analysis of bronchoalveolar lavage fluid (BALF) from critically ill COVID-19 patients revealed upregulation of inflammatory cytokine signatures, indicating an influx of active inflammatory macrophages in the airways [7,8]. Macrophages mediate inflammatory responses following infection via activation of pro-inflammatory responses. Macrophages also migrate to localized infected tissues to mitigate infection. Indeed, an influx of macrophages in the pulmonary tissue of postmortem patients with COVID-19 has been observed [7,9], and virus antigens have been detected in subsets of tissue-resident and lymph node-associated macrophages [8]. These findings implicate macrophages in inflammatory responses during SARS-CoV-2 infection. Therefore, in the current study, we investigated the pro-inflammatory response of SARS-CoV-2-infected human THP-1 macrophages and further elucidated the role of the soluble SARS-CoV-2 spike glycoprotein S1 subunit in inducing such a response in THP-1 macrophages.

## 2. Materials and Methods

Cells: Cell cultures were maintained at 37 °C in a 5% CO_2_ atmosphere. Vero E6 cells (ATCC, Manassas, VA, USA; catalog no. CRL-1586) and HEK293T cells (ATCC, Manassas, VA, USA; catalog no. CRL-3216) were cultured in DMEM medium (ThermoFisher, Waltham, MA, USA; catalog no. 12430062) supplemented with 10% FBS, 100 IU/mL penicillin, and 100 µg/mL streptomycin. Human monocyte-like cells (THP-1 cell line, ATCC, Manassas, VA, USA; catalog no. TIB-202) were cultured in RPMI 1640 medium (ThermoFisher, Waltham, MA, USA; catalog no. 21870076) supplemented with 10% FBS, 10 mM HEPES, 1 mM sodium pyruvate, 50 µM beta-mercaptoethanol, 100 IU/mL penicillin, and 100 µg/mL streptomycin. ACE2-expressing HEK293T cells were generated by transduction with an ACE2-expressing lentivirus made from plasmid pLVX-ACE2 (a gift from Dr. Edward Campbell, Loyola University Chicago) and the packaging plasmids described below. Stably transduced cells were selected with puromycin (0.25 μg/mL) and ACE2 expression was monitored by Western blot analysis.

Virus: SARS-CoV-2 isolate USA-WA1/2020 (BEI resources catalog no. NR-52281) was propagated in Vero E6 cells to generate a virus stock with a titer of 1.76 × 10^6^ 50% tissue culture infective dose (TCID_50_)/mL. All SARS-CoV-2 titrations were performed by TCID_50_ assay on Vero E6 cells, and titers were calculated by the method of Reed and Muench. Work with infectious virus was performed in biosafety cabinets within a biosafety containment level 3 facility. Personnel wore powered air purifying respirators (MAXAIR Systems, Irvine, CA, USA) during all procedures.

Pseudotyped Virus Assays: Pseudotyped lentiviral particles were generated following published protocols [10] by transfection of HEK293T cells using the TransIT-293 reagent (Mirus Bio, Madison, WI, USA) with the following plasmids: pHAGE-CMV-Luc2-IRES-ZsGreen-W (BEI Resources, NR-52516), HDM-Hgpm2 (BEI Resources, NR-52517), HDM-tat1b (BEI Resources, NR-52518), and pRC-CMV-Rev1b (BEI Resources, NR-52519), along with a viral glycoprotein-expressing plasmid: pcDNA3.1-SARS-CoV-S, pcDNA3.1-SARS-CoV-2-S (D614G), or pCAGGS-VSV-G (KeraFast, Boston, MA, USA). The SARS spike-expressing plasmids were provided by Dr. Thomas Gallagher, Loyola University Chicago [11]. Pseudotyped lentiviral particles were used to inoculate HEK-ACE2 cells (HEK293T cells stably expressing human ACE2) or THP-1 cells. GFP expression was monitored by fluorescence microscopy and luminescence was measured at various time points using the Bright-Glo reagent (Promega, Madison, WI, USA) and a FLUOstar Optima microplate reader (BMG LabTech, Ortenberg, Germany).

THP-1 cell infection: THP-1 monocyte-like cells were seeded in plastic dishes in the presence of phorbol 12-myristate 13-acetate (PMA, 100 ng/mL) to induce differentiation into macrophages. After 24 h of incubation, undifferentiated cells were washed away and the attached differentiated macrophages were incubated in fresh media without PMA for an additional 24 h. THP-1 cells were either mock infected with supernatants from non-infected Vero E6 cells or inoculated at an MOI of 0.5 for 1 h at 37 °C with virus stock generated in Vero E6 cells. Cells were washed once with PBS and incubated at 37 °C in complete media for the indicated times. Vero E6 cells were seeded and incubated for 24 h before virus infection at an MOI of 0.1, following the same procedure as with THP-1 cells. After the indicated times, culture supernatants were collected for titration assays (TCID_50_) and RNA was extracted from infected cells using the RNeasy Plus Mini Kit (Qiagen, Germantown, MD, USA; catalog no. 74134), following the manufacturer’s instructions. RNA extracted from THP-1 cells treated with 100 ng/mL of LPS (Invivogen, San Diego, CA, USA; catalog tlrl-eklps;) for 4 h was used as a positive control.

THP-1 cell treatment with S1 purified protein: THP-1 monocyte-like cells were seeded in the presence of phorbol 12-myristate 13-acetate (PMA, 100 ng/mL) and allowed to differentiate as described above. Differentiated macrophages were treated with 8 nM (0.6 μg/mL) recombinant soluble SARS-CoV-2 spike S1 protein subunit purified from HEK293 cells (SinoBiological, Wayne, PA, USA; catalog no. 40591-V08H-B) or an equivalent volume of vehicle control for the specified times. Cell culture supernatants were collected for ELISA assays, and RNA was extracted from infected cells using Trizol (ThermoFisher, Waltham, MA, USA; catalog no. 15596026) following the manufacturer’s instructions.

Reverse Transcription Quantitative PCR (RT-qPCR): For cellular genes, total RNA (500 ng) was used for cDNA synthesis using a High-Capacity cDNA Reverse Transcription Kit (Applied Biosystems, Waltham, MA, USA; catalog no. 4368814) following the manufacturer’s instructions. Approximately 20 ng of cDNA was used as a template for qPCR reactions using SSOAdvanced Universal SYBR Green Supermix, following the manufacturer’s instructions (BioRad, Hercules, CA, USA; catalog no. 1725271). The following primers were used for the detection of cellular genes by qPCR:
GAPDHFw: 5′-ACAACTTTGGTATCGTGGAAGG-3′;
Rv: 5′-GCCATCACGCCACAGTTTC-3′TNF-αFw: 5′-CCTCTCTCTAATCAGCCCTCTG-3′;
Rv: 5′-GAGGACCTGGGAGTAGATGAG-3′IL6Fw: 5′-ACTCACCTCTTCAGAACGAATTG-3′;
Rv: 5′-CCATCTTTGGAAGGTTCAGGTTG-3′IFN-γFw: 5′-TCGGTAACTGACTTGAATGTCCA-3′;
Rv: 5′-TCGCTTCCCTGTTTTAGCTGC-3′IFN-βFw: 5′-GCTTGGATTCCTACAAAGAAGCA-3′;
Rv: 5′-ATAGATGGTCAATGCGGCGTC-3′CXCL10Fw: 5′-GTGGCATTCAAGGAGTACCTC-3′;
Rv: 5′-GCCTTCGATTCTGGATTCAGACA-3′

qPCR reactions were performed in a CFX96 Touch Real-Time PCR Detection System (BioRad, Hercules, CA, USA). The relative gene expression of the target genes was determined using the average Ct for technical replicates normalized to GAPDH. The fold change over mock-infected cells was determined using the 2^−ΔΔCt^ method.

For the quantification of viral RNA, total RNA (250 ng or 500 ng) was used for cDNA synthesis using SuperScript IV VILO Master Mix (Invitrogen, catalog no. 11756050) according to the manufacturer’s instructions. cDNA was diluted to 1:10 and used as a template for qPCR reactions using PowerUp SYBR Green Master Mix (Applied Biosystems, Waltham, MA, USA; catalog no. A25742). Primers for detection of SARS-CoV-2 genes [12] N (Fw: 5′-CAATGCTGCAATCGTGCTAC-3′; Rv: 5′-GTTGCGACTACGTGATGAGG-3′) and S (Fw: 5′-GCTGGTGCTGCAGCTTATTA-3′; Rv: 5′-AGGGTCAAGTGCACAGTCTA-3′) were used for qPCR, along with the GAPDH primers listed above for THP-1 cell samples or actin primers (Fw: 5′-AAGGATTCATATGTGGGCGATG-3′; Rv: 5′-TCTCCATGTCGTCCCAGTTGGT-3′) for Vero cell samples. qPCR reactions were performed using a StepOnePlus Real-Time PCR System (Applied Biosystems, Waltham, MA, USA). Ct values were determined using Design and Analysis 2.5.0 (Applied Biosystems, Waltham, MA, USA) and normalized to GAPDH or Actin Ct values using 2^−ΔCt^.

Enzyme-linked immunosorbent assay (ELISA): Human TNF Alpha Uncoated ELISA Kit (ThermoFisher, Waltham, MA, USA; catalog no. 88-7346) was used to determine secretion of TNF-α in culture supernatants of cells treated with the SARS-CoV-2 S1 subunit. The cytokine concentration was calculated according to the manufacturer’s instructions.

Western blot analysis: To evaluate the expression of ACE2, HEK-ACE2 (HEK293T cells stably expressing human ACE2), THP-1, and Vero cells were lysed using 1%-Triton X-100 (pH 7.4) and EDTA-free protease inhibitor cocktail (Roche Diagnostics, Indianapolis, IN, USA; catalog no. 11836170001) in PBS. Cell lysates were subjected to SDS-PAGE, and separated proteins were transferred onto 0.2 μm nitrocellulose membrane (ThermoFisher, Waltham, MA, USA) and blotted with antibodies specific to ACE2 (Abcam, Waltham, MA, USA; catalog no. ab108252) and actin (Bethyl Laboratories, Montgomery, TX, USA; catalog no. A300-491A). Immunoblots were developed using Western Lightning ECL Pro (Perkin Elmer, Waltham, MA, USA; catalog no. NEL120E001EA) and imaged in Bio-Rad Chemidoc XRS+ (Bio-Rad, Hercules, CA, USA).

Statistical analysis: Two-way ANOVA, adjusted by Sidak’s multiple comparison test, was performed to evaluate the relative expression from the RT-qPCR and ELISA data from three experimental groups compared at multiple time points. A *p* value of <0.05 was considered significant for all statistical tests. All statistical tests were performed using GraphPad Prism v6.01 (San Diego, CA, USA).

## 3. Results

### 3.1. Soluble Glycosylated SARS-CoV-2 Spike Protein S1 Subunit Induces Pro-Inflammatory Response in Human THP-1 Macrophages

It has been reported previously that the purified SARS-CoV-2 trimeric spike (S) glycoprotein produced in mammalian cells [13] and the purified S1 subunit produced in *E. coli* [14] activate pro-inflammatory responses in macrophages. However, these two forms of S proteins do not reflect the physiologically relevant S protein that is generated during SARS-CoV-2 infection of mammalian cells. Prefusion trimeric S is cleaved by cellular proteases [15] that dissociate the S1 subunit during virion assembly [16] or after engagement of the ACE2 receptor [17]. Therefore, trimeric S is transient on the surface of virions, and as reported earlier [13], constructs designed to stabilize this conformation do not reflect the dynamic state of the S glycoprotein and subunits in contact with cells. Importantly, in such a scenario, dissociated S1 may remain engaged to cell receptors and stimulate yet-undefined effects. Likewise, purified S1 derived from non-mammalian sources such as *E. coli* [14] does not reflect a physiologically relevant form of the S1 protein, since S1 is glycosylated at numerous positions [18] that mediate functions such as shielding of viral epitopes. Glycosylation patterns are not recapitulated in proteins purified from *E. coli,* and, thus, non-glycosylated S1 produced in *E. coli* may not reproduce the biological effects of SARS-CoV-2 S1.

To clarify the role of S1 in activating a pro-inflammatory response, we tested whether glycosylated, soluble SARS-CoV-2 S1 purified from mammalian cells induced the expression of pro-inflammatory and antiviral cytokines in human THP-1 macrophages. We treated THP-1 macrophages with purified S1 protein to evaluate the expression of the pro-inflammatory cytokines TNF-α, CXCL10, and IFN-γ [19,20] due to their association with hyperinflammation in SARS and COVID-19 patients, as well as the antiviral cytokine IFN-β, since it restricts SARS-CoV-2 infection [21]. While S1 did not induce gene expression of IFN-β (Figure 1A) or IFN-γ (Figure 1B) in THP-1 macrophages, expression of proinflammatory TNF-α (Figure 1C) and CXCL10 (Figure 1D) was upregulated following exposure to S1. TNF-α was significantly upregulated, by 30-fold, at 4 h post-treatment with S1 and remained higher than vehicle treatment, though not statistically significant, 16 h after treatment (Figure 1C). CXCL10 expression was consistently upregulated by 3- to 8-fold in THP-1 macrophages exposed to S1 up to 16 h post-treatment (Figure 1D). Although macrophages respond to IFN-γ by producing CXCL10, they are not substantial sources of IFN-γ [22], which is mostly produced by lymphocytes to recruit macrophages to infection sites [23]. Thus, SARS-CoV-2 S1 upregulated CXCL10 independently of IFN-γ, similarly to other stimuli such as LPS [24] and TNF-α [25].

We further examined the release of TNF-α from THP-1 macrophages treated with the SARS-CoV-2 S1 subunit. Treatment with S1 induced secretion of TNF-α by macrophages at 4 h and 8 h post-treatment (Figure 1E). These results demonstrated that soluble, glycosylated S1 alone suffices to activate a pro-inflammatory response in human macrophages independently of full-length S proteins, S-trimers, and virus infection. Non-glycosylated S1 purified from *E. coli* induced TNF-α secretion in murine macrophages [14]. Non-glycosylated S1 [14] may expose sites that trigger a pro-inflammatory response, which are cryptic in the glycosylated S1 protein. Thus, we demonstrated that physiologically relevant glycosylated S1 derived from mammalian cells has pro-inflammatory activity in human macrophages.

### 3.2. Human THP-1 Macrophages Do Not Support SARS-CoV-2 Entry or Productive Viral Replication

Since SARS-CoV-2 S1 induced a pro-inflammatory response in macrophages independently of virus infection or replication (Figure 1), we next evaluated viral replication in THP-1 macrophages inoculated with SARS-CoV-2. As expected, infection with SARS-CoV-2 in susceptible Vero E6 cells led to an exponential increase in viral nucleocapsid (N) (Figure 2A) and S (Figure 2B) RNA. Instead of an exponential increase consistent with active replication, the levels of N (Figure 2C) and S (Figure 2D) RNAs diminished over time in THP-1 macrophages. While some viral RNA was detected at 0 hpi, corresponding to 1 h post-virus adsorption, expression did not increase over time for either the N or S genes. Virus replication and release of infectious progeny was determined by TCID_50_ assays in supernatants from SARS-CoV-2-infected cells to corroborate the viral RNA findings. While infected Vero E6 cells supported the robust release of infectious virions due to productive replication, the infectious virus from infected THP-1 cells did not increase and became undetectable after 8 h post-infection (Figure 2E). Finally, THP-1 macrophages did not develop cytopathic effects (CPE) following SARS-CoV-2 infection, whereas Vero E6 cells displayed progressive cell rounding and monolayer damage (Figure 2F). These results together indicate that THP-1 human macrophages do not support productive replication of SARS-CoV-2. Our study is in accord with reports showing non-productive replication of SARS-CoV-2 in human monocyte-derived macrophages and DCs [26,27,28].

We then investigated whether S-mediated entry of SARS-CoV-2 occurred in THP-1 macrophages. Pseudotyped lentiviruses were created that harbored the vesicular stomatitis virus (VSV) G protein (VSV-G), SARS-CoV-1 S, or SARS-CoV-2 S glycoproteins. We first confirmed pseudotyped virus entry into HEK293T cells stably expressing ACE2 (HEK-ACE2). Our results show that all three pseudotyped viruses entered HEK-ACE2 cells (Figure 2G). Next, we transduced THP-1 macrophages with the pseudotyped viruses. Although the VSV-G pseudotyped lentiviruses entered THP-1 macrophages, there was no entry by the pseudotyped viruses bearing the S proteins of SARS-CoV-1 or SARS-CoV-2 (Figure 2H). The S proteins of both SARS-CoV-1 and SARS-CoV-2 viruses use ACE2 as a receptor for cell entry. Thus, we analyzed ACE2 expression in THP-1 cells. The ACE2 protein was expressed by virus-susceptible Vero cells and HEK-ACE2 cells, but was not detected in THP-1 cells (Figure 2I). These results suggest that the S protein containing S1 facilitates the entry of SARS-CoV-2 into ACE2-expressing cells, but not that of THP-1 macrophages, which lack ACE2 expression. However, purified S1 proteins triggered a pro-inflammatory response in THP-1 macrophages (Figure 1). This reflects a non-functional role of virus-associated S in THP-1 entry, whereas the virus-independent soluble S1 protein confers pro-inflammatory activity/function on THP-1 macrophages.

### 3.3. Low-Grade Pro-Inflammatory Response in SARS-CoV-2 Exposed Human THP-1 Macrophages in the Absence of Productive Viral Replication

Despite the lack of replication or S-mediated entry of SARS-CoV-2, the expression of TNF-α (Figure 3A) and CXCL10 (Figure 3B) in THP-1 macrophages inoculated with SARS-CoV-2 was significantly upregulated by 2- and 3-fold at 4 and 24 hpi, respectively. Similar to the response to S1 (Figure 1), SARS-CoV-2 infection did not induce antiviral IFN-β (Figure 3C) or IFN-γ (Figure 3D) expression in THP-1 macrophages, whereas LPS upregulated both cytokines by 80- and 3-fold, respectively. Since both TNF-α and CXCL10 are key pro-inflammatory cytokines, our results suggest a low-grade pro-inflammatory response in THP-1 macrophages exposed to SARS-CoV-2 in the absence of productive virus replication.

## 4. Discussion

Macrophages promote inflammation by producing pro-inflammatory cytokines and chemokines. The role of human macrophages in SARS-CoV-2 infection remains unclear despite their function as pro-inflammatory cells and their contribution to immune dysregulation. Our study shows that although human THP-1 macrophages do not support productive virus replication or S-mediated entry, exposure to SARS-CoV-2 upregulates the expression of pro-inflammatory mediators linked to generalized hyperinflammation in COVID-19 patients [4]. Similarly, SARS-CoV-1 has been shown to induce inflammatory responses in macrophages without viral replication [29]. Moreover, we show that the soluble, glycosylated S1 subunit produced in mammalian cells is sufficient to induce this response in the absence of viral infection. However, antiviral cytokines were not induced, suggesting that S1-activated macrophages would contribute to hyperinflammation, rather than effective antiviral responses. This reflects the reduced or delayed type I interferon (IFN) response and hyperinflammatory response observed in severe cases of COVID-19 [30].

Our study identifies the SARS-CoV-2 soluble, glycosylated S1 subunit as a viral factor involved in the activation of pro-inflammatory responses in human macrophages. Therefore, formation of S trimers or even full-length S is not required to induce this response. The soluble glycosylated S1 subunit triggers a pro-inflammatory response in non-infected macrophages, and, therefore, the interaction of the extracellular S1 subunit is sufficient to induce this response independently of viral infection. Shedding of dissociated S1 has been shown during expression of the full-length S protein on the surface of pseudotyped viruses [17] and when S constructs are expressed in mammalian cells [31]. These studies showed reduced shedding of S1 in S proteins with the D614G mutation [17,31]. Furthermore, S1 has been detected in the plasma of patients with severe COVID-19 during the acute phase of disease, and full-length S has been detected in post-acute phase patients [32]. Thus, we envision that extracellular, soluble S1 released from infected lung epithelial cells may interact with uninfected macrophages and trigger a pro-inflammatory response that contributes to disease pathology (Figure 4). Others have proposed that S1 activates leukocytes by interaction with TLR2 [33] or TLR4 [14], possibly through the galectin-fold domain in S1 [34].

Exposure of macrophages to S1 did not induce any expression of type I or type II IFN, and slightly lower expression levels were observed compared to the vehicle at the earliest time point, 4 h after treatment (Figure 1). This agrees with other studies that have observed increased expression of inflammatory cytokines, but not IFNs, by macrophages stimulated with S1 [33]. One study also found that treatment of cells from bronchoalveolar lavage of rhesus macaques with SARS-CoV-2 S1 results in the inhibition of both basal and poly I:C-induced levels of type I IFN mRNAs [35]. It is unknown whether SARS-CoV-2 or S1 can actively down-regulate the expression of IFN-γ.

In addition, while SARS-CoV-2 infection in THP-1 macrophages does not lead to CPE (indicative of cell death) or productive viral replication, the low-grade inflammatory response remains upregulated for at least 24 h after exposure to SARS-CoV-2 (Figure 3B). Therefore, virus-infected epithelial cells, or S1-activated macrophages, may become sources of pro-inflammatory mediators contributing to hyperinflammation in COVID-19 patients. Our study provides evidence for a contribution of human macrophages to inflammation-associated immunopathology of SARS-CoV-2 by two possible mechanisms (Figure 4)—macrophage activation by (1) the S protein on virions that are released from infected lung epithelial cells, and (2) extracellular soluble S1 proteins released from infected lung epithelial cells and virions.

## Figures and Tables

**Figure 1 viruses-15-00754-f001:**
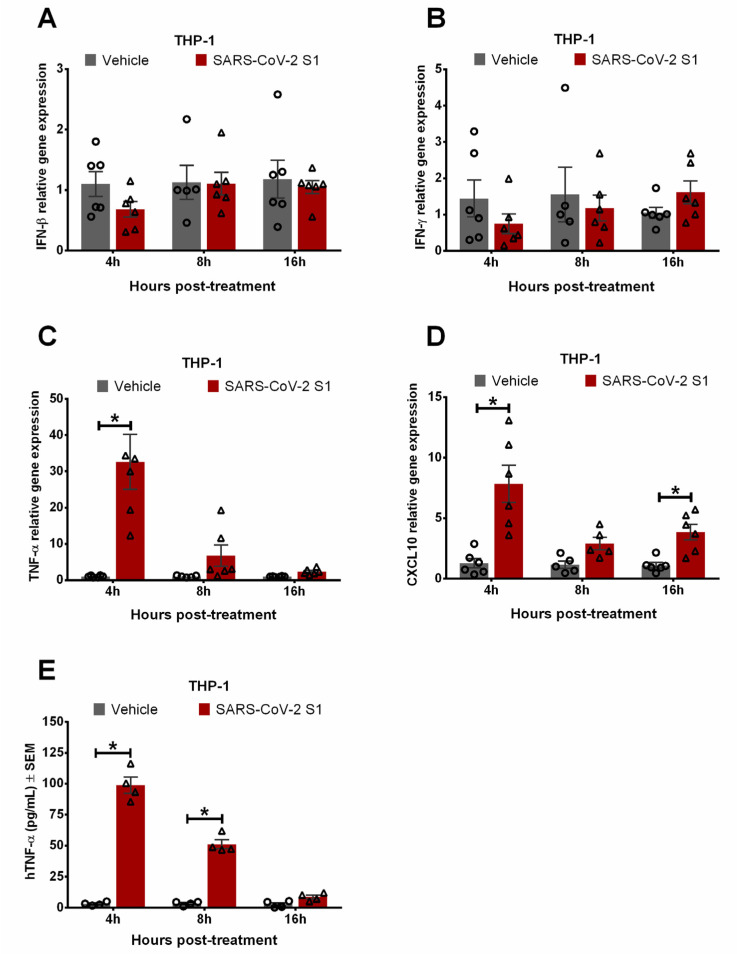
SARS-CoV-2 soluble, glycosylated spike protein S1 subunit (S1) induces a pro-inflammatory response in human THP-1 macrophages. THP-1 cells were treated with purified, recombinant, soluble S1 protein (0.6 μg/mL, 8 nM) or vehicle for the indicated times. RT-qPCR was used to quantify the relative gene expression of IFN-β (**A**), IFN-γ (**B**), TNF-α (**C**), and CXCL10 (**D**). (**E**) Secretion of TNF-α was determined by ELISA assays in supernatants from THP-1 cells treated with purified, recombinant, soluble S1 proteins. Error bars denote the standard error of the mean (SEM) from 3 biologically independent experiments. * *p* < 0.05, determined by two-way ANOVA adjusted by Sidak’s multiple comparison test.

**Figure 2 viruses-15-00754-f002:**
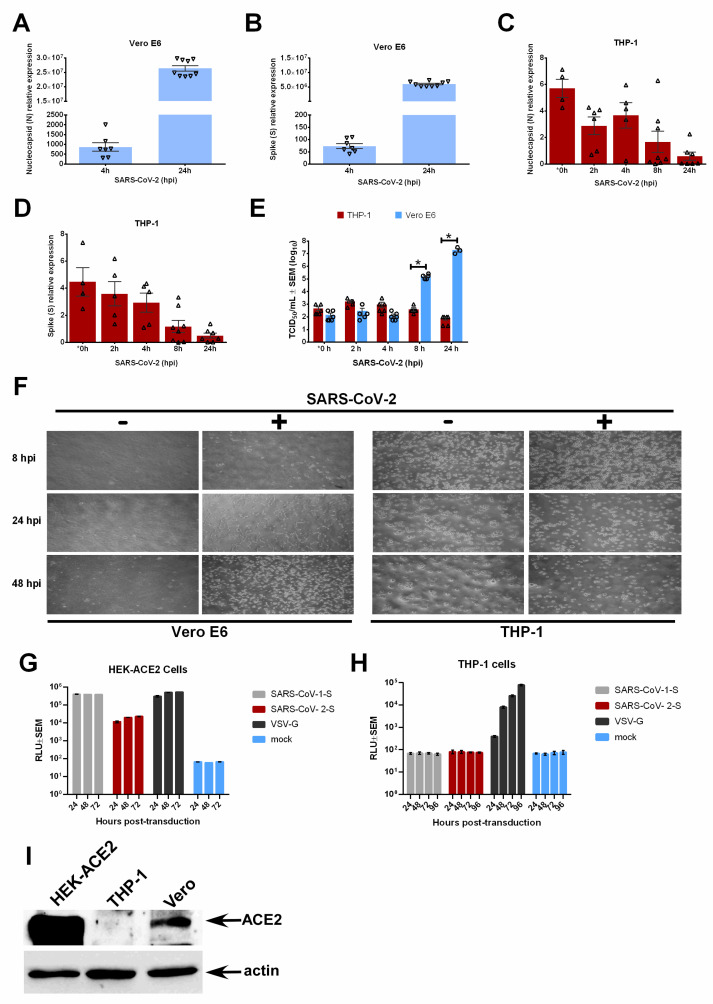
RT-qPCR was used to detect SARS-CoV-2 nucleocapsid or N (**A**) and Spike or S (**B**) viral genes in SARS-CoV-2-infected Vero E6 cells. RT-qPCR was used to detect N (**C**) and S (**D**) viral genes in SARS-CoV-2-infected human THP-1 macrophages. (**E**) Culture supernatants from SARS-CoV-2-infected human THP-1 macrophages and Vero E6 cells were analyzed by TCID_50_ assay to determine infectious virus production. (**F**) Bright field microscopy photographs of Vero E6 (MOI = 0.1) and THP-1 (MOI = 0.5) macrophages infected with SARS-CoV-2 for the indicated times. Luminescence (RLU) was measured to determine transduction of HEK-ACE2 cells (HEK293T cells stably expressing human ACE2) (**G**) and THP-1 macrophages (**H**) by luciferase-expressing lentiviruses pseudotyped with VSV glycoprotein (VSV-G), SARS-CoV-1 S protein, or SARS-CoV-2 S protein. (**I**) ACE2 expression was analyzed in HEK-ACE2, THP-1, and Vero cells by Western blotting with ACE2 and actin (loading control) antibodies. Error bars denote the standard error of the mean (SEM) from 2 to 3 biologically independent experiments. hpi = hours post-infection. * *p* < 0.05 was determined by two-way ANOVA adjusted by Sidak’s multiple comparison test. * 0 h timepoint data were collected 1 h post-virus adsorption.

**Figure 3 viruses-15-00754-f003:**
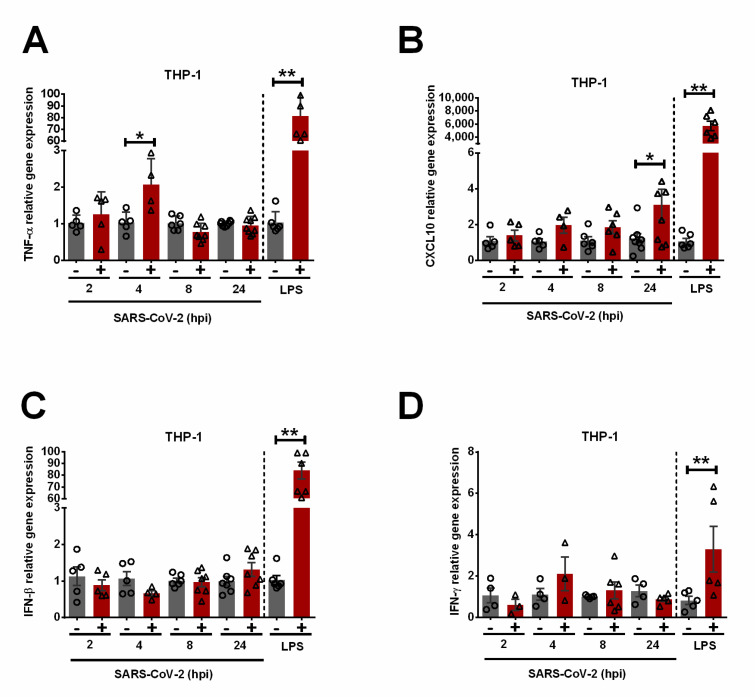
Low-grade pro-inflammatory response in SARS-CoV-2-exposed human THP-1 macrophages in the absence of productive virus replication. RT-qPCR was used to detect relative gene expression of TNF-α (**A**), CXCL10 (**B**), IFN-β (**C**), and IFN-γ (**D**) in SARS-CoV-2-inoculated THP-1 macrophages. LPS-treated macrophages (100 ng/mL, 4 h) were used as positive controls. Error bars denote the standard error of the mean (SEM) from 2 to 3 biologically independent experiments. hpi = hours post-infection. * *p* < 0.05, ** *p* < 0.01 was determined by two-way ANOVA adjusted by Sidak’s multiple comparison test.

**Figure 4 viruses-15-00754-f004:**
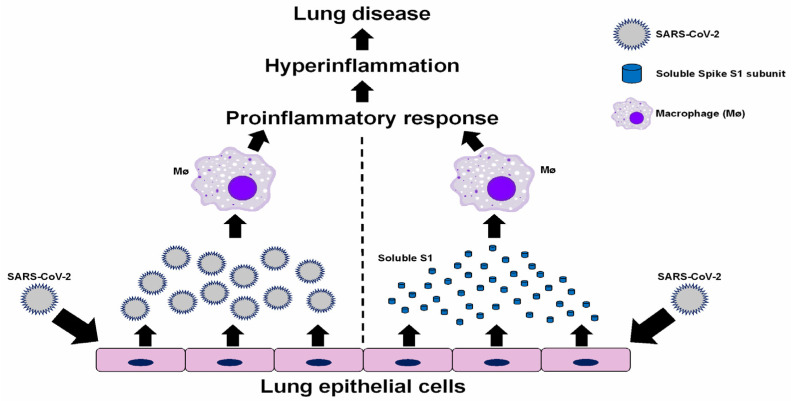
Proposed contribution of SARS-CoV-2 and soluble S1 to inducing inflammatory responses by macrophages. SARS-CoV-2 virions and soluble S1 proteins released from productively infected lung epithelial cells trigger proinflammatory responses by macrophages, which then contribute to hyperinflammation and lung disease associated with COVID-19.

## Data Availability

Data reported in the manuscript are available from the corresponding author.

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
