# Peer review of "Proinflammatory Responses in SARS-CoV-2 and Soluble Spike Glycoprotein S1 Subunit Activated Human Macrophages"

_viruses, 2023, doi:10.3390/v15030754_

Round 1

Reviewer 1 Report

This main manuscript focused on the proinflammatory signaling of macrophages in response to the soluble S1 spike protein of SARS-CoV-2. This study was one of the first to look at the expression of proinflammatory cytokines in macrophages specifically. The manuscript was well written and the scientific logic is sound. However, there are a few concerns that should be addressed.

#1 LPS was used as a positive control to elicit an inflammatory response. However, LPS is a bacterial lipid that elicits a bacterial-specific response (i.e., through bacterial-specific toll-like receptors). It would be best to include a viral-specific positive control in these experiments to better compare the SARS-CoV-2 S1-specific response.

#2 In Figure 1A and B, primarily at 4 hours, the S1 protein decreases INF-β and IFN-γ compared to the vehicle. This result is unexpected. Is there evidence that the S1 protein inhibits IFN1 expression?

#3 In Figure 1C, it is stated that there is a 2-fold upregulation of TNFα expression at 16 hours. However, the figure does not clearly show this upregulation. Is this upregulation significant?

#4 On line 213, it is stated that THP-1 macrophages do not support productive replication of SARS-CoV-2. Is the lack of productive replication in these cells due to the virus itself or the macrophages actively fighting off the infection? Do THP-1 cells express ACE2?

#5 In the Discussion, it is proposed that S1 interacts with TLR4 to produce the proinflammatory cytokines observed in this study. However, S1 is a glycoprotein and glycoproteins have been shown to interact with receptor for AGEs (RAGE) resulting in the production of proinflammatory cytokines (i.e., TNFα and IL-6). Could S1 be interacting with RAGE in addition to TLR4 to drive the production of proinflammatory cytokines? Additionally, it would be interesting to block TLRs and RAGE and see if the proinflammatory cytokines are still produced and which receptors are responsible for the production of the cytokines.

#6 On line 292 and the conclusion in Figure 4, it is stated that a possible mechanism for the inflammation-associated immunopathology of SARS-CoV-2 is the virus activation of human macrophages. However, the data in Figure 3 shown that viral-infected THP-1 cells did not result in any significant increase in proinflammatory cytokines. Therefore, the conclusion made on line 292 and Figure 4 does not match the data shown.

Reviewer 2 Report

In this manuscript, the authors investigate the inflammation of THP-1 cell triggered by SARS-CoV-2 virus and the spike glycoprotein subunit 1. They found that virus infection in macrophages does not lead to cell death or productive viral replication, low-grade inflammatory response remains upregulated for at least 24 hours after exposure to virus. Therefore, virus-infected cells, or S1-activated macrophages may become sources of pro-inflammatory mediators contributing to hyperinflammation in COVID-19 patients.

From the literature, there are several similar studies which concluded that SARS-CoV-2 virus or the spike proteins can induce pro-inflammation responses in THP-1 cells. In addition, the possible mechanisms are also investigated.

For instance:

https://www.frontiersin.org/articles/10.3389/fimmu.2021.683800/full

https://www.ncbi.nlm.nih.gov/pmc/articles/PMC7987013/

https://www.nature.com/articles/s42003-021-02983-5

It is suggested to revise the manuscript and add more mechanism based studies.

Author Response

Point 1: It is suggested to revise the manuscript and add more mechanism based studies.

Response 1: Thank you for providing additional references, which we incorporated into the manuscript. Our study was submitted as a brief report. While mechanism-based studies are important, they are beyond the scope of this study.

Reviewer 3 Report

The authors Kim et al investigated pro-inflammatory response in SARS-CoV-2 infected human THP1 macrophages. They also demonstrated that the spike glycoprotein S1 subunit of SARS-CoV-2 induces THP1 macrophages.

Here are my comments.

1. the manuscript is based on two methods. qPCR and ELISA, I would suggest including western blot analysis for proinflammatory cytokines.

2. Fig 2. F demonstrating the microscopy pictures of Vero and THP1 cells are not clear. I would strongly suggest working on them.

In Conclusion, this study is conclusive given the suggestions above.

Author Response

Point 1: the manuscript is based on two methods. qPCR and ELISA, I would suggest including western blot analysis for proinflammatory cytokines.

Response 1: We used ELISA as it is the standard in the field for quantifying cytokines. ELISA is more sensitive and quantitative than western blot analysis.

Point 2: Fig 2. F demonstrating the microscopy pictures of Vero and THP1 cells are not clear. I would strongly suggest working on them.

Response 2: We have increased the size of the microscopy pictures in Fig. 2F to better demonstrate the change in cell morphology and monolayer in SARS-CoV-2 infected Vero E6, but not THP-1 cells.

Round 2

Reviewer 1 Report

Thank you for addressing my questions and concerns. I feel like the revised and added information greatly clarifies this manuscript. At this time I feel like this manuscript is suitable for publication.

Thank you.